

# A priority experience replay actor-critic algorithm using self-attention mechanism for strategy optimization of discrete problems

Yuezhongyi Sun and Boyu Yang

School of Computer Science and Technology, Harbin University of Science and Technology, Harbin, Heilongjiang Province, China

## ABSTRACT

In the dynamic field of deep reinforcement learning, the self-attention mechanism has been increasingly recognized. Nevertheless, its application in discrete problem domains has been relatively limited, presenting complex optimization challenges. This article introduces a pioneering deep reinforcement learning algorithm, termed Attention-based Actor-Critic with Priority Experience Replay (A2CPER). A2CPER combines the strengths of self-attention mechanisms with the Actor-Critic framework and prioritized experience replay to enhance policy formulation for discrete problems. The algorithm's architecture features dual networks within the Actor-Critic model—the Actor formulates action policies and the Critic evaluates state values to judge the quality of policies. The incorporation of target networks aids in stabilizing network optimization. Moreover, the addition of self-attention mechanisms bolsters the policy network's capability to focus on critical information, while priority experience replay promotes training stability and reduces correlation among training samples. Empirical experiments on discrete action problems validate A2CPER's adeptness at policy optimization, marking significant performance improvements across tasks. In summary, A2CPER highlights the viability of self-attention mechanisms in reinforcement learning, presenting a robust framework for discrete problem-solving and potential applicability in complex decision-making scenarios.

## INTRODUCTION

Deep reinforcement learning (DRL) remains a central subfield within artificial intelligence, continuously attracting research interest (*Ladosz et al., 2022*). Central to DRL is the goal of empowering intelligent agents with optimal decision-making capabilities in dynamically evolving environments (*Li, 2023*; *Diallo & Padilla, 2018*). These agents actively engage with their surroundings, with deep neural networks guiding their decisions and actions (*Osband et al., 2016*). Among the various algorithms in DRL, the Actor-Critic (AC) algorithm is recognized as a prominent and effective approach (*Kubo, Chalmers & Luczak, 2022*).

Corresponding author
Boyu Yang,
2220410118@stu.hrbust.edu.cn

This quintessential methodology in DRL leverages the strengths of policy networks (Actors) and value function networks (Critics), providing a robust framework for addressing the field's multifaceted challenges (*Yang et al., 2021*). In the AC algorithm, the policy network develops action strategies, while the value function network estimates the states or state-action pairs, guiding the iterative improvement of policies (*Wei et al., 2022*). This synergy endows the AC algorithm with significant adaptability and generalization capabilities in learning tasks (*Kapoutsis et al., 2023*).

However, traditional AC algorithms have encountered certain limitations, especially when dealing with discrete problems. These limitations primarily arise from the policy network's insufficient modeling capacity to capture inter-state correlations and the inherent inefficiencies of training within large state spaces (*Ciosek et al., 2019*). To overcome these challenges, our study integrates the self-attention mechanism (*Shaw, Uszkoreit & Vaswani, 2018*) into the AC algorithm to enhance the representational capacity and improve the policy network's learning efficacy.

Originally a powerful tool in natural language processing, the self-attention mechanism has shown exceptional versatility across various data types and tasks (*Gou et al., 2022*). Its core concept allows the model to dynamically prioritize different aspects of the input data, thus facilitating a deeper understanding of essential information. By incorporating the self-attention mechanism, our goal is to strengthen the AC algorithm, enabling it to better adapt to discrete problems and enhance the policy's generalization capacity (*Niu, Zhong & Yu, 2021*).

In addition to the self-attention mechanism, there are many excellent algorithm ideas worth learning from, such as COA (Catfish Optimization Algorithm), ROA (Raven Optimization Algorithm), and DHHO (Dynamic Harris Hawks Optimization). Each of these algorithms introduces unique strategies and mechanisms that can be synergistically integrated with DRL approaches to tackle complex problem-solving scenarios more effectively.

The Crayfish Optimization Algorithm (COA) (*Jia et al., 2023*), inspired by the hunting behavior and dynamic grouping of crayfish, excels in exploring vast and complex search spaces rapidly, making it particularly useful for environments where the state space explodes in dimensionality. This algorithm enhances the efficiency of exploring new states, potentially reducing the time required for an intelligent agent to learn optimal policies.

Similarly, the Remora Optimization Algorithm (ROA) (*Jia, Peng & Lang, 2021*), drawing inspiration from the intelligence and adaptability of remoras, provides mechanisms for dynamic adaptation to changing environments. This adaptability is crucial for DRL applications where the environment's dynamics are unpredictable, requiring algorithms that can adjust their strategies in real time.

Lastly, the DHHO algorithm (*Jia et al., 2019*), based on the cooperative behavior and tactical precision of Harris hawks, offers refined search capabilities and convergence properties. Its dynamic adjustment of exploration and exploitation phases makes DHHO exceptionally suitable for balancing between diversification and intensification strategies in policy optimization (*Heidari et al., 2019*).

By incorporating elements from these algorithms—COA's rapid exploration, ROA's adaptive capabilities, and DHHO's balanced search dynamics—into the AC framework, we can significantly enhance the model's performance. These integrations allow for a more nuanced understanding and exploitation of the environment, ultimately leading to more robust and effective policy development.

This article's main contributions include the proposal of an AC algorithm augmented by the self-attention mechanism for optimizing policies in discrete problem domains (*Wang et al., 2016*). We empirically demonstrate its superior performance on discrete tasks compared to traditional AC algorithms. Additionally, we explore potential applications and future research directions of the self-attention mechanism within the realm of reinforcement learning.

## RESEARCH BACKGROUND

### Actor-critic algorithm

Actor-Critic Algorithm (*Konda & Tsitsiklis, 1999*): In the realm of reinforcement learning, the Actor-Critic (AC) algorithm is recognized as a fundamental methodology (*Siyao et al., 2022*; *Lowe et al., 2017*). It divides the intelligent agent into two core components: the Actor, which is the policy network and the Critic, which is the value function network. The Actor's role is to design action policies and guide decision-making. It continually refines its parameters through ongoing interactions with the environment, aiming to enhance action selection efficiency and maximize rewards. In contrast, the Critic estimates the values of states or state-action pairs and evaluates the quality of the policy by striving to develop an accurate value function.

The synergy between the Actor and the Critic is of paramount importance within the AC algorithm (*Yang et al., 2021*). The Actor relies on the Critic's value estimations to inform its policy improvement efforts, favoring actions that are more likely to yield significant rewards. As the training process progresses, the agent interacts with the environment by observing states, executing actions, and collecting rewards. The Actor uses the actions suggested by the current policy, while the Critic uses the environment's reward signals to evaluate state values (*Wang, Zeng & Shang, 2023*). In this dynamic interaction, the Actor adjusts the policy to maximize expected cumulative rewards, and the Critic refines the value function, aiming to minimize estimation errors. This collaborative process drives the AC algorithm forward (*Jia & Zhou, 2022*), enabling iterative improvements in policy performance that align with the environmental demands and the complexities of the task. The update formula for the value function network (Critic) is as follows:

$$L(\theta) = E\left\{ \left[ Q(s, a; \theta_{Critic}) - (r + \gamma V(s'; \theta_{Critic}))^2 \right] \right\} \qquad (1)$$

Among them: ( $Qs$, $a$; $\theta_{Critic}$): The Q-value function predicted by the Critic for the current state-action pair (s, a), parameterized by $\theta_{(theta)}$.

$r + \gamma V(s'; \theta_{Critic})$: The target value, which is the sum of:

r: The immediate reward after taking action in state s.

$\gamma V(s'; \theta_{\text{Critic}})$: The discounted value of the next state s, as estimated by the Critic. $\gamma_{(\text{gamma})}$ is the discount factor, which quantifies the difference in importance between future rewards and immediate rewards. $V(s'; \theta_{\text{Critic}})$ is the value function for the next state $s'$, also parameterized by $\theta$.

The loss function $L(\theta)$ aims to adjust the parameters $\theta$ of the Critic network to minimize the difference between the predicted Q-values and the target values (the sum of the immediate reward and the discounted value of the next state). Minimizing this loss results in a critique that better approximates the true Q-values of the policy being evaluated.

The operation proceeds as follows:

For each sample (or batch of samples), compute the Critic's Q-value prediction $Q(s, a; \theta_{\text{Critic}})$ for the current state-action pair.

Calculate the target value $r + \gamma V(s'; \theta_{\text{Critic}})$ for the next state, combining the immediate reward and the value of the next state discounted by the factor $\gamma$.

Compute the square of the difference between the Q-value prediction and the target value.

Take the expectation (average over a batch or samples) of the squared differences, which gives us the value of the loss function $L(\theta)$.

Perform gradient descent or another optimization algorithm to adjust $\theta$ in a way that minimizes $L(\theta)$.

In summary, the Actor-Critic (AC) algorithm utilizes the strengths of policy search and value function estimation by facilitating cooperation between the Actor and Critic networks. This cooperative dynamic allows the AC algorithm to perform robustly across various reinforcement learning tasks, efficiently optimizing policies in complex and high-dimensional environments.

However, AC algorithms face several challenges (*Fujimoto, Hoof & Meger, 2018*). Training instability is a prime issue, especially with complex tasks and high-dimensional state spaces, where policy updates can dramatically shift expected outcomes, introducing volatility and potential bias. Additionally, AC algorithms often require many samples to train the policy and value function networks effectively, making them resource-intensive for extensive training environments. The algorithms also heavily depend on the precision of the value function estimations; inaccuracies here can negatively affect policy improvements. Moreover, adapting AC algorithms to discrete action spaces typically necessitates specialized optimizations.

Our research seeks to incorporate the self-attention mechanism, a tool that has gained prominence in deep learning, to improve the performance of the AC algorithm. This mechanism allows models to dynamically concentrate on relevant information, thus improving the efficiency of information utilization. Integrating self-attention into the AC framework aims to bolster the policy network's ability to handle high-dimensional states and intricate tasks more effectively.

We also plan to implement priority experience replay to enhance training efficiency and stability. Furthermore, we will apply techniques such as target network updates, gradient

clipping, and loss function optimization to significantly boost the AC algorithm's generalization and robustness.

## Self-attention mechanism

In the context of the A2CPER algorithm introduced in this article, the primary foundation of the self-attention mechanism revolves around the utilization of the QKV (Query, Key, and Value) methodology (*Wang et al., 2022*). This approach comprises three key elements:

Query: The Query, whether in the form of a vector or matrix, serves as a tool to identify specific points of interest within the data. Its crucial role lies in governing the calculation of attention weights. Essentially, the Query acts as a directive, indicating which information deserves heightened attention in the current context (*Ding, Han & Guo, 2021*). Key: The Key, similarly represented as a vector or matrix, functions as a comparative element regarding both the Query and the Value. It plays a pivotal role in assessing the significance and relevance of each piece of information. Value: The Value, often represented as a vector or matrix, encapsulates the factual information, typically derived from the input or the hidden state of the current time step. This information represents the content that is intended to be transmitted to the output (*Zhang et al., 2021*).

The fundamental principle underpinning the attention mechanism is the assignment of varying weights to the Value contingent upon the degree of matching between the Query and the Key. In this manner, the mechanism effectively determines which information merits heightened attention and retention within the prevailing context. Specifically, the calculation of the attention mechanism can be expressed by the following formula:

$$\text{Attention}(Q, K, V) = \text{softmax}\left(\frac{Q * K^T}{\sqrt{d_k}}\right)V \tag{2}$$

Among them: Attention(Q, K, V): This is the attention function that takes three inputs: Q (queries), K (keys), and V (values). The function computes a weighted sum of the values (V), where the weight assigned to each value is determined by the query with the corresponding key.

Q: Queries are a set of vectors that we want to compute attention for. In the context of sequence tasks, these can be the representations of the current word or element for which we are trying to determine context. K: Keys are another set of vectors that are paired with the values. They are used to compute the attention weights. V: Values are the vectors that we want to focus on. Once the weights are determined by the compatibility of Q and K, these weights are applied to the values. Q, K, and V signify queries, keys, and values respectively—these are vectors representing the inputs in our model. The queries embody the current element we seek to elucidate, Keys are paired with values which contain the actual information from the input sequence we want to draw from.

The dot product $Q * K^T$ measures similarity; here, $K^T$ is the transpose of K, facilitating the alignment in dimensions necessary for the product operation. This similarity dictates the degree of 'attention' the queries should allocate to each value.

The resultant scores are scaled down by the factor of $\sqrt{d_k}$, where $d_k$ is the dimensionality of the key vectors. This scaling counters the potential vanishing gradients problem by mitigating the impact of large dot product values which can result from high-dimensional spaces.

Finally, a softmax function is applied, converting the scaled scores into a probabilistic distribution—ensuring the attention scores across the sum of the values to one. This distribution is then utilized to compute a weighted sum of the values, thereby producing an output that signifies the input elements' contextual representation.

In our model development, we specifically employ Xavier initialization, an approach demonstrably effective in optimizing the learning of neural network weights.

This method sets the stage for an enhanced learning trajectory of our attention-based strategies. According to *Xie et al. (2023).* the algorithmic construct for distributing attention weights emphasizes the amplification of operations pivotal to the core tasks of the model. Such prioritization is crucial for magnifying the impact of operations with measurable outcomes, enhancing model performance. Conversely, operations contributing to less critical tasks or associated with suboptimal outcomes receive proportionally diminished attention weights. This stratified allocation of weights is instrumental in situating the initial parameters within a conducive range for rapid convergence, facilitating an expedited and efficient training phase. Our attention reward formula, devised to reflect these principles, is optimized for task-centric weight distribution:

$$\text{Attention Scores} = \frac{(QW_q) * (W_k K^T)}{\text{sqrt}(d_k)} \qquad (3)$$

Among them: $(QW_q)$: This term represents the matrix multiplication of Q (queries) with $W_q$, the weight matrix associated with queries. Q encapsulates the input vectors that the model seeks to elucidate, and $W_q$ is a learnable parameter matrix that transforms the queries into an appropriate space for subsequent operations.

$(W_k K^T)$: Similarly, $W_k$ is the weight matrix corresponding to K (keys) and $W_k K^T$ denotes its transpose. The keys are vectors that, in conjunction with the queries, will determine the attention distribution over the values.

$(QW_q) * (W_k K^T)$: This operation signifies the dot product between the transformed queries and the transposed transformed keys, which calculates the alignment scores between them. In the Transformer model, these scores reflect the extent to which each element of the queries should attend to each element of the keys.

$\text{sqrt}(d_k)$: Here, $d_k$ denotes the dimensionality of the key vectors. The square root of this value is utilized as a scaling factor to moderate the magnitude of the dot product scores, thereby averting potential issues with gradient descent optimization such as gradient vanishing or explosion. This scaling is particularly critical in high-dimensional spaces common in complex models.

The formula can thus be read as: The attention scores are the result of a scaled dot product operation between the transformed queries and keys. These scores are

subsequently used in the softmax step to obtain the final attention weights, which are applied to the values (V) in the attention mechanism.

The linear transformations applied to Q (queries) and K (keys) facilitate the model's ability to project input states into the corresponding query and key spaces. These transformations are instrumental in capturing the correlations between states. The terms $QW_q$ and $W_kK^T$ denote the resultant vectors from the linear transformation of queries and keys, respectively. When multiplied, they yield the attention scores. A pivotal aspect of this optimization framework is the incorporation of learnable weight matrices $W_q$ and $W_k$, enabling the model to adaptively refine the weighting of queries and keys, thus enhancing the calculation of attention scores. This adaptability of the weight matrices allows the model to autonomously fine-tune attention allocations in response to task-specific demands. The factor $\sqrt{d_k}$ serves as a normalizing constant, scaling the attention scores to mitigate against excessively large gradients, thereby preserving numerical stability. Both the SelfAttention layer within the policy network and the value function network are interfaced with a fully connected layer, designated as 'fc1'. This layer is tasked with transforming the input state to match the input dimensionality required by the SelfAttention layer, enabling the model to learn an efficient mapping from states to their self-attention representations.

## Priority experience replay

Recognizing the suboptimal efficiency of the sampling mechanism in Actor-Critic (AC) algorithms, we have integrated a Priority Experience Replay (PER) strategy to enhance sampling efficacy, as suggested by *Schaul et al. (2015)*. Central to the PER methodology is the principle of preferential replay of experience tuples, whereby experiences are assigned priority levels that influence their probability of being replayed during training. High-priority samples, as they possess a greater probability of selection, disproportionately inform the learning process, thereby expediting convergence and augmenting algorithmic efficiency (*Gong et al., 2022*).

To refine this priority-based sampling methodology, we introduced several optimizations:

Priority assignment: Each experience is allocated a priority score that quantifies its perceived relevance to the learning context.

Analysis and weighting: Post initial sampling, an assessment of priority levels is conducted. The resultant priority scores are used to adjust the weighting of experiences within the replay buffer, prioritizing those with greater relevance to the learning process.

Data selection: A judicious selection process is then employed, wherein data with higher weighted priorities are preferentially chosen for training iterations. This ensures that experiences with significant learning value are given precedence in the training regimen.

Employing the PER approach as delineated addresses the inherent inefficiencies of traditional sampling in AC algorithms and propels the model towards swifter convergence

and improved performance outcomes (*Wei et al., 2022*). The formula for calculating priority weights is presented as follows:

$$\text{Priority} = |\text{TD}| + \varepsilon * \text{W} \tag{4}$$

Among them: Priority: This term refers to the importance or relevance assigned to an individual experience tuple, determining the likelihood that it will be replayed during training. A higher priority increases the chance of an experience being selected for replay, thereby exerting a greater influence on the model's updates.

$|\text{TD}|$: The absolute value of the temporal difference $|\text{TD}|$ error. In reinforcement learning, the the $|\text{TD}|$ error represents the difference between the predicted value of a particular state-action pair and the actual reward received plus the predicted value of the subsequent state. It is an indicator of the surprise or unexpectedness of an experience—larger errors imply that the experience has more to teach the agent.

$\varepsilon$: A small positive constant, often referred to as a smoothing term or 'epsilon'. This ensures that no experience has a zero probability of being replayed, thus maintaining exploration and preventing the model from becoming overly deterministic in its sampling.

W: This term represents a weighting function in the priority calculation. It encapsulates additional considerations that might influence the importance of an experience beyond the TD error.

The formula can be explained as follows: The priority of an experience is determined primarily by the magnitude of its TD error, which signifies how much the experience can potentially contribute to the agent's learning. The small constant $\varepsilon$ ensures that all experiences have some non-zero probability of being selected. The term W indicates a supplementary weighting strategy, which could be used to incorporate additional considerations into the priority score.

When applied, this formula equips the learning agent with a strategy to focus on experiences that are likely to yield the most substantial learning progress, as well as maintain a degree of stochasticity and diversity in the experiences it revisits (*Wu et al., 2023*). This balance accelerates learning and helps to ensure that the agent does not become trapped in local optima, a common pitfall in machine learning optimization. The weighting method uses the empirical attributes of the collected samples and weights the error values to reduce the impact of sample randomness on the experiment.

## Target network

To augment the stability and efficacy of the Actor-Critic (AC) algorithm, the A2CPER framework has been endowed with a target network mechanism. This innovation delays the parameter updates within both the Actor and Critic networks, fulfilling two critical objectives: it strengthens the algorithm's stability and reinforces the robustness of the training dynamics (*Miller, Xiang & Kesidis, 2020*).

In the realm of deep reinforcement learning, the target for training is often a moving goalpost, imbued with inherent volatility that can precipitate instability throughout the learning process. Our strategy to counteract this challenge involves the implementation of

staggered parameter updates. Here, the Actor and Critic networks' parameters are recalibrated by the gathered training data, albeit with a strategic temporal offset. While the updates persist in alignment with the trajectory of the training, they are executed in a manner that significantly dampens oscillations and curtails the propagation of erroneous learning signals (*Miller, Xiang & Kesidis, 2020*).

The introduction of a fixed temporal window during which the target network remains static engenders a stable and consistent training target. Such stabilization is instrumental in diminishing the risk of instability and in reducing the prevalence of approximation errors. Collectively, these modifications enhance the performance and dependability of the A2CPER algorithm. The target network update formula is as follows:

$$\theta_{target} = \alpha * \tau * \theta_{current} + (1 - \alpha) * \theta_{target} \tag{5}$$

Among them: $\theta_{target}$: These represent the parameters of the target network, which are used to compute the target values against which the current network's predictions are evaluated.

$\theta_{current}$: These are the parameters of the current network, which are being actively updated through backpropagation in response to the learning signal.

$\alpha$: This is the interpolation parameter, often a small value close to 0. It dictates the rate at which the target network parameters are updated. A smaller value of $\alpha$ means the target parameters will change more slowly, thereby smoothing out learning and enhancing stability.

The formula can be interpreted as follows: The updated target network parameters ($\theta_{target}$) are a weighted combination of the previous target network parameters and the current network's parameters. The weighting is controlled by $\alpha$, allowing for a controlled rate of change. This results in a controlled progression of target values, which helps the learning process by providing stable, slowly moving targets. This is especially important in complex environments where the learning signal can be noisy or the optimization landscape can be rugged.

The weighting factor we use is a value between 0 and 1. It is used to control the weight distribution between the current network parameters ($\theta_{current}$) and the target network parameters ($\theta_{target}$). Specifically: when $\alpha = 0$, the update in the formula only depends on the target network parameters, the current network parameters have no effect, so the entire update is a basic soft update. When $\alpha = 1$, the update in the formula only depends on the current network parameters and the target network parameters have no effect, which is equivalent to a complete hard update. When $0 < \alpha < 1$, the update is affected by a trade-off between the two network parameters, that is, both the current network and the target network participate in the update. The purpose of the improved target network update formula in this article is to achieve more flexible parameter updates by mixing current network parameters and target network parameters. By adjusting the value of $\alpha$, you can control the weight distribution between the current network and the target network, thereby balancing the stability of training and the speed of learning.

## Gradient clipping

In addition to these mechanisms, gradient clipping is employed as a stabilizing technique in deep reinforcement learning. Its primary purpose is to curtail the magnitude of the gradient, thereby averting the detrimental impacts of gradient explosions and bolstering training stability. Within the A2CPER framework, the estimation of the policy gradient is computed as follows.

$$\nabla J(\theta) \approx \frac{1}{N} \sum_{i=1}^{N} \nabla_\theta \log\left(\pi\left(\frac{a_i}{s_i}\right)\right) \cdot R_i \tag{6}$$

Among them: $\nabla J(\theta)$: This represents the gradient of the performance measure J concerning the policy parameters $\theta$. The performance measure J is typically the expected cumulative reward. The gradient indicates the direction in which the parameters should be adjusted to increase the expected reward.

$\frac{1}{N}$: This term represents the average over N sampled episodes. Averaging across multiple samples is crucial to reduce the variance of the gradient estimate and improve the stability of learning.

$\sum_{i=1}^{N} \nabla_\theta$: This is the summation of the N episodes, each contributing to the estimate of the gradient.

$\nabla_\theta \log\left(\pi\left(\frac{a_i}{s_i}\right)\right)$: The logarithm of the policy $\pi$, which outputs the probability of taking action $a_i$ given state $s_i$, is differentiated concerning $\theta$. The log probability is used because it is more mathematically tractable and because its gradient is more informative than the probability itself.

$R_i$: This is the cumulative reward received in the i-th episode. It serves as a scalar value that modulates the parameter update—episodes with higher rewards will have a larger impact on the direction of the gradient.

The policy gradient formula as a whole represents an expectation over the product of the log probability of the policy's actions and the cumulative reward. By taking steps in the direction of this gradient, the policy's parameters are adjusted to increase the likelihood of actions that lead to higher rewards.

The Critic network uses a policy gradient to calculate the gradient. The calculation formula is as follows:

$$\nabla J(\theta) = \sum \left[\nabla_\theta \log\left(\pi\left(\frac{a}{s}\right)\right) \cdot A(s, a)\right] \tag{7}$$

Among them: $\nabla J(\theta)$: This denotes the gradient of the performance objective J concerning the policy parameters $\theta$. Objective J is typically defined as the expected return, and this gradient points in the direction of increasing the expected return.

The summation symbol indicates that the calculation involves a sum over all state-action pairs $(s, a)$ encountered in the environment, weighted by the gradient of the log-probability of the policy $\pi\left(\frac{a}{s}\right)$ concerning the policy parameters $\theta$.

$\nabla_\theta \log\left(\pi\left(\frac{a}{s}\right)\right)$: This is the gradient of the log probability of selecting action a in state s, according to the current policy $\pi$. The use of the logarithm simplifies the gradient calculation due to the log-derivative trick, and it can lead to more stable updates.

A(s, a): This represents the advantage function at state s and action a, which estimates how much better it is to take a particular action in the state s compared to the average action at that state. The advantage function is central to variance reduction techniques in policy gradient methods.

The formula suggests that the parameter updates of the policy are proportional to the product of the advantage function and the gradient of the log probability of the policy. By scaling the updates with the advantage, the algorithm focuses on increasing the probability of actions that yield higher-than-average returns, effectively guiding the policy towards more profitable behaviors.

The function of gradient clipping is to trim the gradient of the model to ensure that the range of the gradient is within an appropriate threshold to prevent training instability caused by excessive gradient values. Gradient explosion may lead to large updates of parameters, making the training process unable to converge. The purpose of adding gradient clipping is that if the norm of the sampling calculation gradient exceeds the set threshold ($\max_{\text{grad}_{\text{norm}}}$), the gradient will be scaled so that its norm does not exceed the threshold. This can avoid gradient explosion and maintain the stability of training. The gradient clipping formula is as follows:

$$g_{\text{clipped}} = \text{clip}\left(g, -\max_{\text{grad}_{\text{norm}}}, \max_{\text{grad}_{\text{norm}}}\right) \tag{8}$$

Among them: $g_{\text{clipped}}$: This represents the clipped gradient, which is the modified gradient after the clipping operation has been applied.

clip: This is the clipping function that limits the value of the gradient to a defined range. It takes the original gradient and a specified clipping threshold as inputs.

g: The original gradient computed concerning the model's loss function.

$\max_{\text{grad}_{\text{norm}}}$: This term defines the maximum norm allowed for the gradient. If the norm of the gradient exceeds this value, the gradient will be scaled down to meet this threshold.

The formula indicates that if the norm of the gradient g is greater than $\max_{\text{grad}_{\text{norm}}}$, then the gradient is scaled back to this maximum allowable norm. The purpose of this operation is to ensure the updated gradient does not exceed a magnitude that could destabilize the learning process. By capping the gradient, we can maintain control over the optimization trajectory, making it less likely to experience erratic updates, which is especially beneficial in scenarios with highly non-linear objective functions.

## Loss strategy optimization

The loss function constitutes a cornerstone in the optimization framework of Actor-Critic (AC) algorithms, and within the scope of A2CPER, it is bifurcated into two essential constituents: the policy loss and the value function loss. The policy loss, frequently termed the strategy loss, is instrumental in refining the Actor's network. It achieves this by endeavoring to amplify the expected cumulative reward, relying fundamentally on the log

probability of the selected actions as inferred from the policy network's probabilistic outputs.

This log probability is juxtaposed with the advantage function, which delineates the expected improvement of the chosen action over the baseline strategy. The discrepancy thus evaluated serves as an indicator of the Actor's performance, guiding the optimization process. The computational expression for the strategy's performance function is delineated as follows:

$$J(\theta) = E\left[\sum_{t=0}^{T} \tau^t \frac{R_t}{\pi_\theta}\right] \tag{9}$$

Among them: $J(\theta)$: The performance objective function of the policy, which depends on the policy parameters $\theta$. It represents the expected return, which the learning algorithm aims to maximize.

E: The expectation operator, which indicates that $J(\theta)$ is the expected value of the sum within the brackets. This expectation is taken over the distribution of trajectories ($\tau$) induced by the current policy $\pi_\theta$.

$\sum_{t=0}^{T} 0$: This is the summation of all timesteps from 0 to T, where T could be the terminal timestep of an episode or an arbitrary truncation point for the calculation.

$\tau^t$: This term represents a discount factor raised to the power of the timestep t, which reduces the weight of rewards received at later timesteps, encapsulating the concept of temporal preference where immediate rewards are generally preferred over distant ones.

$R_t$: The reward received at timestep t. In reinforcement learning, the goal is often to maximize the cumulative reward over time.

$\pi_\theta$: The policy function, which gives the probability of taking an action given the current state and policy parameters $\theta$.

The term $\frac{R_t}{\pi_\theta}$ seems to be missing the action taken at timestep t, as the policy probability normally appears in the denominator of such an expression. Typically, this would look like $\frac{R_t}{\pi_\theta(\alpha_t|s_t)}$, representing the probability of taking action $\alpha_t$ in state $s_t$ according to the policy.

The objective function $J(\theta)$ is therefore a sum over all timesteps of discounted rewards weighted by the reciprocal of the action probabilities under the policy. This form suggests an importance sampling approach, where returns are weighted inversely by their probability under the policy, though the exact interpretation may vary depending on the context provided by the accompanying text or formulation.

In the optimization landscape of Actor-Critic algorithms, our methodology enhances the precision of action selection assessment, which is gauged against the baseline of average scenarios. This is complemented by the value function loss, which is integral to the Critic network's refinement. This loss is aimed at reducing the mean squared error, thereby fine-tuning the value function network's accuracy.

To advance our optimization strategy for the loss function, we have implemented a weighted loss mechanism. This mechanism is sensitive to the relative weights of successive

loss values. When a newly computed loss value carries a weight exceeding that of its predecessor, we meticulously document the outcome of this secondary computation. If the scenario does not present a higher weight than the extant one, the current loss value is retained as a benchmark for parameter adjustments. This tactic is designed to guide the trajectory of parameter adjustments, keeping them closely aligned with the benchmarked results and minimizing variance.

This nuanced optimization technique substantively augments the fidelity of our training regimen, curtailing the error margins inherent in loss value computations.

## EXPERIMENTAL SETUPS AND MODELS

### Experimental setup for CartPole-v1

In our experimental work, we leveraged the discrete reinforcement learning model, CartPole-v1, as documented by *Kumar (2020)*. CartPole-v1 is a canonical testbed from the OpenAI Gym suite, extensively adopted for the development and validation of reinforcement learning algorithms. The environment challenges an agent to balance a pole on a mobile cart, necessitating the application of lateral forces to preclude the pole from tipping over. The state space of the environment is defined by four continuous variables: the cart's horizontal position and velocity, alongside the pole's angle and angular velocity. These variables provide the agent with the necessary situational awareness to make informed decisions.

The action space in CartPole-v1 is binary, permitting the agent to exert force to either the left or the right of the cart. The simplicity of the reward system, which assigns a positive increment for each time step the pole remains upright within designated bounds, belies the complexity of the task. An episode terminates, and the cumulative reward is assessed, once the pole's angle surpasses a critical inclination or the cart traverses beyond set spatial limits.

CartPole-v1 distinguishes itself from its predecessor by demanding a protracted balance duration, thereby serving as a rigorous and insightful challenge for evaluating the efficacy of reinforcement learning algorithms.

### Mathematical modeling of CartPole-v1

**State Variables and Dynamics:**

The state of the system at any given time t can be represented by a vector containing the following variables:

$x(t)$: Cart position.

$\dot{x}(t)$: Cart velocity.

$\theta(t)$: Pole angle (concerning the vertical).

$\dot{\theta}(t)$: Pole angular velocity.

The equations of motion for the CartPole system are derived from the physics of an inverted pendulum on a cart. They include gravitational forces, friction, and the force applied by the agent. These can be formulated using the Euler-Lagrange equations or Newtonian mechanics, leading to a set of differential equations.

**Action Space:**

The agent can take one of two actions at any time t:

a(t) = 0: Apply force to the left.

a(t) = 1: Apply force to the right.

**Reward Function:**

The agent receives a reward of +1 for every time step that the pole remains upright, and the episode ends if the pole falls beyond a certain angle or the cart moves out of the allowed boundary.

In mathematical terms, the reward at time t is:

R(t) = 1, if termination conditions are not met.

R(t) = 0, if termination conditions are met (episode ends).

**Termination Conditions:**

$|\theta(t)| >$ threshold angle

$|x(t)| >$ boundary limit

## Experimental setup for Acrobot-v1

The Acrobot-v1 is a classic control task environment in the reinforcement learning domain, provided by the OpenAI Gym interface. The Acrobot is a two-link, two-joint pendulum with the links connected end-to-end and only the second joint actuated. Initially, both links point downwards. The goal is to swing the end of the lower link up to a given height by applying torque on the second joint.

## Mathematical modeling of Acrobot-v1

**State Space:**

The state s of the system at time t can be represented as:

$s(t) = [\cos(\theta_1), \sin(\theta_1), \cos(\theta_2), \sin(\theta_2), \theta_1, \theta_2]$

where $\theta_1$ and $\theta_2$ are the angles of the first and second joints relative to the vertical, and $\theta_1$ and $\theta_2$ are their respective angular velocities.

**Equations of Motion:**

The motion of the Acrobat is governed by a set of non-linear differential equations derived from Lagrangian mechanics:

$\frac{d}{dt} \cdot \frac{\partial L}{\partial \dot{\theta}} - \frac{\partial L}{\partial \theta} = \tau$ where L is the Lagrangian of the system, representing the difference between kinetic and potential energies, and $\tau$ is the torque applied to the second joint.

**Action Space:**

The action space consists of a discrete set, A, where the agent selects the torque to apply: $A = \{-1, 0, +1\}$.

**Reward Function:**

The reward function R(s, a) gives a reward of $-1$ at each timestep unless the terminal conditions are met, encouraging the agent to swing the end of the lower link to the desired height as quickly as possible.

**Objective:**

The objective is to find a policy $\pi$ that maximizes the cumulative reward over an episode.

## Experimental setup for MountainCar-v0

MountainCar-v0 is a reinforcement learning environment from the OpenAI Gym library. In this environment, an underpowered car is situated between two hills. The objective is to drive up the mountain on the right; however, the car's engine is not strong enough to ascend the mountain in a direct route. Therefore, the car must learn to leverage potential energy by driving back and forth to build up enough momentum to reach the goal at the top of the rightmost hill.

## Mathematical modeling of MountainCar-v0

**State Space:**

The state of the car can be described by a two-dimensional vector s(t):

$$s(t) = [x(t),\ \dot{x}(t)]$$

where x(t) is the position of the car on a one-dimensional track, ranging from −1.2 to 0.6, and $\dot{x}(t)$ is the velocity of the car, ranging from −0.07 to 0.07.

**Dynamics equations:**

The dynamics of the car are captured by the following difference equations, which express how the position and velocity of the car change over time:

$$x(t+1) = x(t) + \dot{x}(t)$$
$$\dot{x}(t+1) = \dot{x}(t) + 0.001 \cdot a(t) - 0.0025 \cdot \cos(3x(t))$$

where a(t) is the action force applied at time t.

**Action space:**

The action space is discrete, and the agent can choose one of three actions at any time:

a(t) = 0: Apply full throttle backward (accelerate to the left).

a(t) = 1: Apply no throttle (zero acceleration).

a(t) = 2: Apply full throttle forward (accelerate to the right).

**Reward function:**

At each time step, the reward function R(s, a) is −1, unless the car reaches the target position (x ≥ 0.5), at which point the episode ends.

**Objective:**

The goal is to find a policy $\pi$ that maximizes cumulative rewards, which equates to the car reaching the target position as quickly as possible.

# EXPERIMENTAL RESULTS AND ANALYSIS

## Experiments of CartPole-v1

The experiments conducted in this study were meticulously controlled to ensure the consistency of hyperparameters and to adhere to optimal principles across all experimental aspects (*Eberding, 2022*). We evaluated a suite of reinforcement learning algorithms,

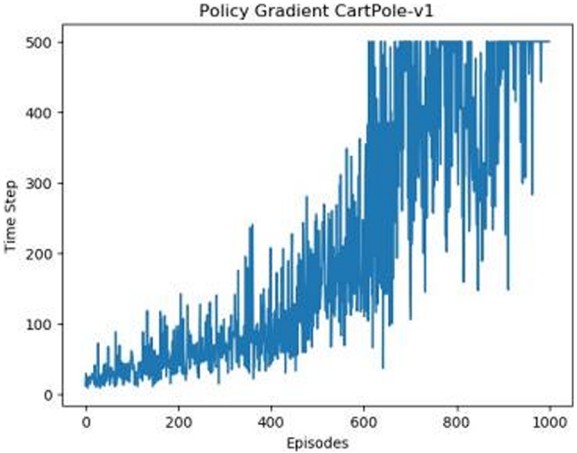

**Figure 1 Policy chart in Experiment 1.**

including the DQN, Policy Gradient, AC, ACER, and the novel A2CPER algorithm, using a comparative approach. The hyperparameters involved in the experiment are provided in the appendix. The selection is based on extensive experience and experimental verification.

Our comparisons were carried out from three distinct perspectives: the overall number of time steps during which the cart maintained balance, the average duration for which the cart sustained balance, and the mean reward value acquired by the cart. To maintain rigor, we executed 1,000 simulation experiments, each with five different random seeds, and computed averages for comparative analysis.

Figures 1–5 in our experiments illustrates the total time steps during which the cart successfully maintained balance. Algorithms within the AC framework reach the target time step of 500 earlier than other algorithms as a whole. Typically, these algorithms exhibit a peak in performance after approximately 100 to 200 iterations. In contrast, other algorithms tend to achieve their first peak performance after around 200 iterations. This discrepancy highlights the AC algorithm's proficiency in initial data collection, attributed to the superior data analysis capabilities of its Critic network. The A2CPER algorithm, proposed in this article, can reach its first peak performance around 100 iterations. This accelerated progress can be attributed to the priority experience replay strategy, which effectively reduces the utilization of irrelevant sampling data and promotes learning from high-priority data, thus expediting the attainment of peak performance.

Additionally, when inspecting the performance of each algorithm, we observed fluctuations during experiments. The degree of fluctuation corresponds to the number of data points in the table. Notably, the A2CPER algorithm exhibited minimal data fluctuations due to the incorporation of the self-attention mechanism. This addition significantly enhanced algorithm stability by prioritizing attention to data fluctuations. Large fluctuations often result from excessive updates to neural network parameters, a challenge effectively mitigated by the A2CPER algorithm through the introduction of delayed parameter updates in target networks.

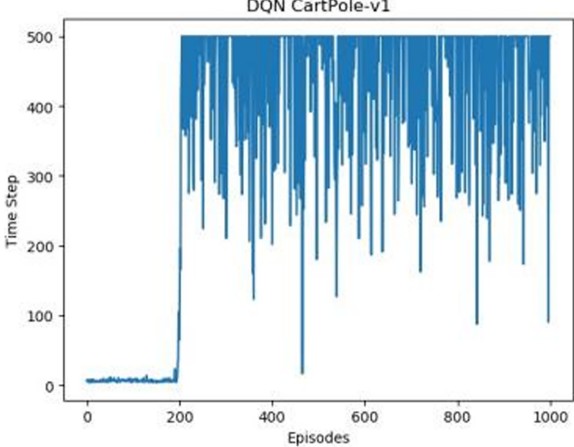

**Figure 2  DQN's graph in Experiment 1.**               

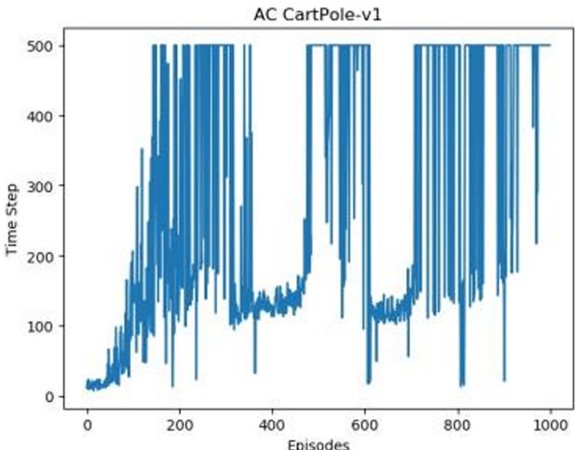

**Figure 3  The graph of AC in Experiment 1.**           

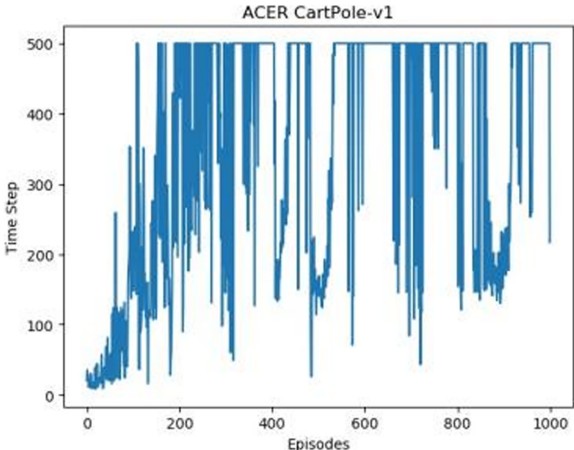

**Figure 4  ACER's graph in Experiment 1.**              

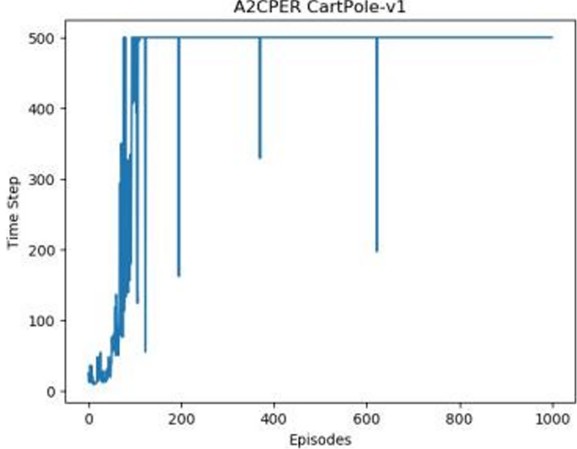

**Figure 5** A2CPER's chart in Experiment 1.     

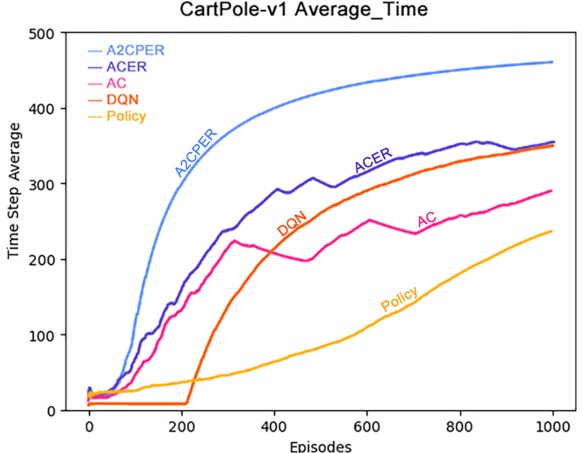

**Figure 6** Comparison chart of average time values of five algorithms in Experiment 1.

Lastly, we observed that algorithms utilizing the AC framework experienced significantly lower training levels than other algorithms during the middle and later stages of the experiments. Subsequently, their performance normalized after approximately 100 iterations. This phenomenon is primarily attributed to gradient-related issues within the loss function. A2CPER optimizes the loss function by incorporating gradient clipping and a weighted loss calculation approach, effectively addressing this challenge.

Overall, our experiments shed light on the strengths of the A2CPER algorithm, particularly in terms of rapid convergence, stability, and enhanced training efficiency.

The numbers in the picture represent: The title represents the algorithm and experimental environment used to experiment; the abscissa is the number of steps in the experiment, in times; the ordinate is the time for the experimental car to maintain balance at each step, in milliseconds.

Figure 6 presents the average duration for which the cart successfully maintained balance. Notably, the A2CPER algorithm exhibited a significantly higher average duration

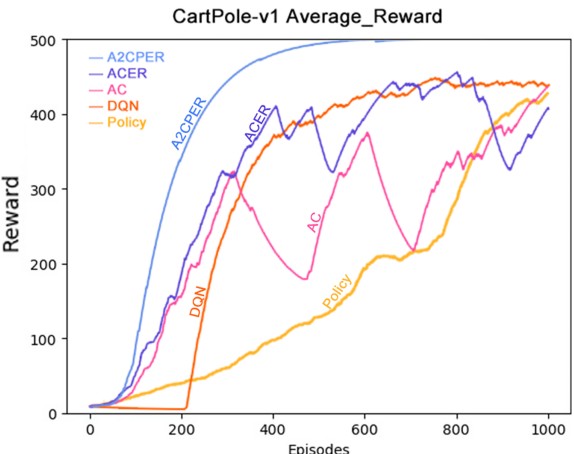

**Figure 7 Comparison chart of average reward values of five algorithms in Experiment 1.**

compared to other algorithms, underscoring the effectiveness of the optimizations it introduced. Conversely, other algorithms displayed varying degrees of shortcomings in this aspect.

During the phase where the average values declined, evident fluctuations were observed across all algorithms. It is worth noting that ACER's average performance closely paralleled that of the DQN algorithm, albeit still falling slightly short. The primary reason for this variance can be attributed to ACER's inability to effectively manage parameter optimization delays within the target network, resulting in excessive fluctuations.

These findings emphasize the advantages conferred by the A2CPER algorithm, particularly in terms of achieving higher average durations for maintaining balance, demonstrating its superior performance and stability compared to alternative algorithms.

Data representation in the figure: the title is the experimental environment and experimental objectives, the abscissa is the number of steps in the experiment, the ordinate is the average time for the car to maintain balance, the upper left corner of the figure represents different algorithms, which are represented by different colors. Pictures can be drawn in a color-blind-friendly way. The algorithm for drawing pictures adopts the web direct drawing method provided by the Data-Driven Documents website.

Figure 7 depicts the average reward value obtained by the cart. To mitigate the influence of occasional fluctuations stemming from individual training scenarios, we have reduced the weighting of each training's reward value. This adjustment ensures that sporadic fluctuations have a limited impact on the overall dataset.

As demonstrated in the figure, the A2CPER algorithm achieves an average reward value peak of around 400 iterations and maintains this level for an extended period. This observation underscores the effectiveness of our optimizations in addressing discrete problems similar to the one presented in this study. Furthermore, the curve exhibits remarkable smoothness, indicative of effective control over fluctuations. These findings reinforce the A2CPER algorithm's capacity for consistent and stable performance in addressing challenging reinforcement learning tasks.

Data representation in the figure: the title is the experimental environment and experimental objectives, the abscissa is the number of steps in the experiment, the ordinate is the average reward for the car to maintain balance, the upper left corner of the figure represents different algorithms, which are represented by different colors. Pictures can be drawn in a color-blind-friendly way. The algorithm for drawing pictures adopts the web direct drawing method provided by the Data-Driven Documents website.

## Results of Cartpole-v1

The A2CPER algorithm's performance gauged through both the duration of the cart's balance and the accumulated reward, underscores its optimization advantages within this discrete setting compared to other algorithms. It has enhanced the Actor-Critic (AC) algorithm by refining the data sampling process, rectified the non-prioritized data collection in the Actor-Critic with Experience Replay (ACER) algorithm, alleviated the prolonged data sampling duration associated with the Deep Q-Network (DQN), and expedited the convergence velocity that typically constrains the Policy Gradient approach.

In addressing the CartPole-v1 environment, the A2CPER algorithm exhibits a time complexity of $O(d \cdot p)$, where d represents the dimensionality of the input data and $p$ denotes the number of parameters in the neural network. The algorithm's runtime decreases progressively with training iterations, as demonstrated in the balance duration graph, ultimately exceeding 500 ms. The convergence rate is also depicted in the graph; while other algorithms typically converge after about 300 iterations, the A2CPER algorithm achieves convergence in just 100 iterations.

Specifically for the CartPole-v1 environment, the implementation of the A2CPER algorithm significantly enhances the performance of the cart. This improvement stems firstly from the introduction of the self-attention mechanism, which focuses on the key objective during the cart's strategy formulation to maintain balance. This mechanism effectively prevents the incorporation of excessive and irrelevant factors into the learning process. Additionally, the priority experience replay technique provides a replay space where the cart needs only to retrieve experiences from the buffer based on priority weights for each training session, filtering out a substantial amount of meaningless data. This approach also includes the use of target networks to moderate parameter updates and gradient clipping to prevent gradient explosion. Through multiple training cycles, our model demonstrated progressively improved performance, ultimately learning the optimal strategy to maximize cumulative rewards.

## Experiments of Acrobot-v1

Following the experiments conducted in the CartPole-v1 environment, we extended our investigation to the Acrobot-v1 environment. The core methodology of the experiments mirrored that of the CartPole-v1, focusing on analyzing the time taken to reach the target height and the acquisition of rewards, with comparisons based on calculated averages. The hyperparameters used in the experiments are detailed in the appendix, selected as the most suitable based on extensive testing.

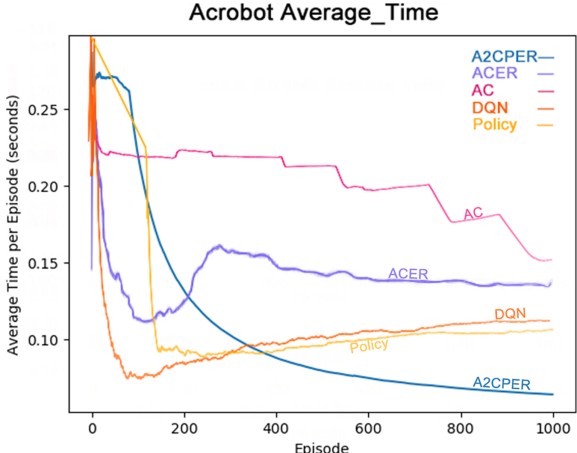

**Figure 8 Comparison chart of the average time of the five algorithms in Experiment 2.**

Figure 8 illustrates the average time taken by five algorithms—A2CPER, ACER, AC, DQN, and Policy—to enable the Acrobot agent to reach the target height. It is evident from the graph that, except for the A2CPER algorithm, all other algorithms exhibited various degrees of increase in average times during their trials, indicating a regression in learning. In contrast, the A2CPER algorithm consistently showed a decrease in average time without any upward regression, underscoring its superior consistency over other algorithms. This consistency is attributed to the introduction of the priority experience replay mechanism, which prevents significant disparities in the training environment that could lead to negative learning outcomes. This phenomenon is notably observable in ACER, where around 300 training iterations, the algorithm is significantly impacted by 'dirty data' in the experience replay buffer, resulting in negative learning.

Furthermore, the A2CPER algorithm's average time was consistently lower compared to the other algorithms, demonstrating its effectiveness in addressing this discrete problem. This performance benefit is due to the incorporation of the self-attention mechanism and the integration with target networks, which ensure that parameter updates are both targeted and gradual.

The chart provides data on the average duration it takes for the robotic arm to reach the target height, measured in seconds. The x-axis represents the number of experimental steps (in iterations), while the y-axis denotes the average time in seconds. The top right corner of the graph includes a color legend distinguishing the different algorithms. It is essential to ensure that the chart is color-blind friendly, possibly using patterns or distinct color contrasts that are easily distinguishable. The chart could be generated using web-based tools like those provided by Data Driven Documents (D3.js) for direct drawing.

Figure 9 illustrates the average reward obtained by the same algorithms—A2CPER, ACER, AC, DQN, and Policy—in the Acrobot-v1 environment (*Cobbe et al., 2021*). The graph clearly shows that the A2CPER and ACER algorithms perform similarly in terms of reward acquisition, but A2CPER consistently outperforms ACER. Both significantly surpass the AC algorithm in efficiency, highlighting the clear benefits of implementing an

**Figure 9 Comparison chart of average reward values of five algorithms in Experiment 2.**

experience replay mechanism. Like the average time data, the average reward values for the other four algorithms exhibit fluctuations, suggesting that the integration of the attention mechanism and other methods in the A2CPER algorithm effectively minimizes such variability, representing a substantial improvement.

The data displayed in the graph represents the average reward values achieved as the robotic arm reaches the targeted height. The x-axis represents the number of experimental steps (in iterations), and the y-axis represents the average reward value. The lower right corner of the graph differentiates the algorithms by using distinct colors, and the chart is designed to be color-blind friendly, using web-based direct drawing tools from the Data-Driven Documents (D3.js) website.

## Results of Acrobot-v1

The comparative analysis of average time and average reward values demonstrates the superior optimization performance of the A2CPER algorithm in this discrete environment. It not only addresses the stability issues, preventing negative learning but also improves the data collection inefficiencies observed in the AC algorithm and the issues with 'dirty data' in ACER's experience replay buffer. Furthermore, it enhances the stability which was notably lacking in the DQN and Policy algorithms.

In addressing the Acrobot-v1 environment, the time complexity of the A2CPER algorithm is represented as $O(d \cdot p)$, where d is the dimensionality of the input data, and p is the number of parameters in the neural network. The runtime of the algorithm decreases with each training iteration, and performance data provided in the balance duration graph indicate times exceeding 500 ms. The convergence rate is also depicted in the graph, showing that A2CPER achieves effective convergence within about 200 iterations.

Regarding the Acrobot-v1 environment, the implementation of the A2CPER algorithm has significantly enhanced the ability of the robotic arm to address challenges. Initially, the introduction of the self-attention mechanism allows the arm to quickly identify the most effective movement strategies to reach the target height. The priority experience replay

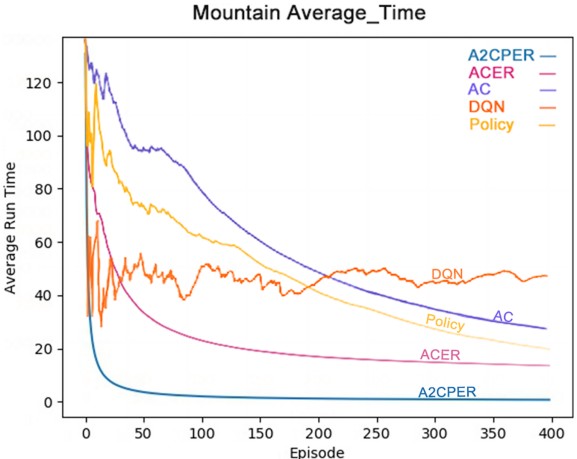

**Figure 10 Comparison chart of the average time of the five algorithms in experiment 3.**

technique employed in the experiments enables the arm to select actions based on their weighted importance and adapt based on the outcomes, effectively filtering out irrelevant data. Additionally, the use of target networks moderates the update of parameters, and gradient clipping is employed to prevent gradient explosions. Through multiple training cycles, our model demonstrated progressively improved performance, ultimately learning the optimal strategy to maximize cumulative rewards.

## Experiments of MountainCar-v0

Finally, to ensure the comprehensiveness of our experiments, we conducted further tests using the MountainCar-v0 environment. Due to the limitations of this environment, where reward values are not effectively observable, we focused our experiments on the time taken for the car to reach the mountaintop, comparing the averages across different algorithms. Each of the five algorithms was trained 400 times to determine which performed best. All hyperparameters used in this experiment are displayed in the appendix section, selected as optimal values after extensive data analysis.

Figure 10 shows the average time it takes for the Policy, DQN, AC, ACER, and A2CPER algorithms to get the MountainCar-v0's car to the target point. From the graph, it is clear that the A2CPER and ACER algorithms exhibit overall stability without significant fluctuations. This indicates that the introduction of the experience replay mechanism significantly enhances the stability of the algorithms. In contrast, the other three algorithms experienced substantial fluctuations, suggesting that excessive extraction of 'dirty data' occurred, leading to these variations.

Furthermore, the final average time for A2CPER approaches 0 milliseconds, while the other four algorithms are around 20 milliseconds, demonstrating the high efficiency of the A2CPER algorithm in dealing with this discrete problem. This efficiency is attributed to the synergy between the self-attention mechanism and multiple methods, culminating in excellent outcomes.

The data displayed shows the experimental environment and objectives, with the x-axis representing the number of experimental steps (in iterations), and the y-axis indicating the time taken for the car to reach its target, measured in milliseconds. Different algorithms are represented by distinct colors in the top right corner of the graph. The visualization is designed to be color-blind friendly, utilizing a web-based direct drawing method provided by the Data-Driven Documents website.

### Results of MountainCar-v0

The comparative analysis of the average time metrics shows that the A2CPER algorithm demonstrates significant optimization advantages in this discrete environment relative to other algorithms. Its most notable impact is the enhancement of algorithmic stability; even in the calculation of average values, which typically do not exhibit large fluctuations, other algorithms showed significant variability, whereas A2CPER maintained nearly zero overall volatility and delivered excellent performance, effectively resolving the car's uphill challenge. It improved the slow convergence issue seen with the AC algorithm, addressed the poor average time results of the ACER algorithm, and stabilized the unusually high fluctuations exhibited by the DQN and policy algorithms.

In managing the MountainCar-v0 environment, the time complexity of the A2CPER algorithm is denoted by $O(d \cdot p)$, where d represents the dimensionality of input data and $p$ denotes the number of parameters in the neural network. The runtime of the algorithm decreases with continued training iterations, achieving about 2 ms toward the end. The convergence rate is also presented in the graph, with A2CPER achieving robust convergence within approximately 20 iterations.

For the MountainCar-v0 environment, the A2CPER algorithm provided a highly effective solution for navigating the car uphill. Initially, the introduction of the self-attention mechanism in the training process allowed the system to quickly discern the most effective movement strategies for reaching the peak. The priority experience replay technique employed during the experiments enabled the robotic arm to select actions based on weights and adjust based on outcomes, filtering out much irrelevant data. This includes the use of target networks to slow down parameter updates and gradient clipping to prevent gradient explosions. Through multiple training cycles, our model demonstrated progressively improved performance, ultimately learning the optimal strategy to maximize cumulative rewards.

## CONCLUSIONS

In a series of experiments with robotic agents solving discrete problems in this study, the advantages of the A2CPER algorithm are evident. First of all, this study effectively avoids unnecessary parameters and ignores irrelevant situations through the introduction of the self-attention mechanism, allowing the agent to quickly acquire useful methods. Secondly, the priority experience playback technology introduced by office research eliminates a large amount of dirty data; even if some data enters the buffer, it will be excluded due to its priority, ensuring that each data batch used by the agent is valid and highly relevant. In addition, this study also made some optimizations. For example, the introduction of the

target network slowed down the speed of parameter updates and prevented large fluctuations caused by rapid changes in parameters. The introduction of gradient clipping ensures that the algorithm remains stable under various conditions and does not cause gradient explosion. Finally, the optimization of the loss value calculation method enables the most effective loss value calculation method to be used in different scenarios. Overall, these properties of this study make the A2CPER algorithm highly effective and worthy of further research and exploration.

### Funding

The authors received no funding for this work.

### Competing Interests

The authors declare that they have no competing interests.

### Author Contributions

- Yuezhongyi Sun conceived and designed the experiments, performed the experiments, analyzed the data, performed the computation work, authored or reviewed drafts of the article, and approved the final draft.
- Boyu Yang conceived and designed the experiments, performed the experiments, analyzed the data, performed the computation work, prepared figures and/or tables, and approved the final draft.

### Data Availability

The data is available at GitHub and Zenodo: - https://github.com/791381705/A2CPER. - yangboyu. (2024). 791381705/A2CPER: A2CPER (Version A2CPER). Zenodo. https://doi.org/10.5281/zenodo.11126762.

### Supplemental Information

Supplemental information for this article can be found online at http://dx.doi.org/10.7717/peerj-cs.2161#supplemental-information.

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
