# Peer review of "A priority experience replay actor-critic algorithm using self-attention mechanism for strategy optimization of discrete problems"

_PeerJ Computer Science, doi:10.7717/peerj-cs.2161_

## Round 0.1 · original submission · Major Revisions

It's clear that a major revision is needed. The authors should focus on improving language clarity, addressing structural issues, providing clearer explanations of the proposed method, conducting additional analyses and comparisons.

Reviewer 1 ·

Basic reporting

The manuscript is well-written and promising. However, I think remediating the following missings will improve the quality of the study.

A flowchart and/or algorithm of the proposed model would help in an easy understanding of the model.

The model was tested with only the CartPole-v1 problem. In order to say that the model is general for discrete problems, it needs to be tested with more problems.

Experimental design

No comment (please see the above)

Validity of the findings

No comment (please see the above)

Reviewer 2 ·

Basic reporting

no comment

Experimental design

In the abstract, it is stated that “The efficacy of the A2CPER algorithm is validated through a series of experiments conducted on discrete action problems.” But only one application problem was discussed in the actual paper.

Validity of the findings

The A2CPER algorithm includes multiple key aspects. In the experimental results, the author need to point out that which one have improved the performance of the compared experiments.

Additional comments

It is suggested that the A2CPER algorithm is illustrsted with structural diagram.

Reviewer 3 ·

Basic reporting

The language of the manual is not clear and understandable.
When the literature is examined, the references are made correctly.
The structure of the Manuscript is generally not fluent and there are breaks in the article.
Comparisons are not enough in the conclusion section of the manuscript. The conclusion section needs to be improved.

Experimental design

The manuscript is suitable for the purpose and scope of the journal. However, I think that the academic aspect is weak in terms of handling the subject.
The manual states, “It introduces an innovative deep reinforcement learning algorithm called A2CPER.”

In this study, the Actor-Critic algorithm is basically discussed. It is then combined with the Priority Experience Replay method. There are deficiencies in explaining the method put forward by the authors. Pseudo codes and flow diagrams should be given to make it more understandable.

Validity of the findings

In experimental studies in Manuscript, CartPole-v1 is used.
CartPole-v1 is a model frequently used in such studies.
Experimental study results are shown by the authors in Figure 2-7. However, each figure must be explained separately.

In order to provide more evidence in experimental results, the study should be presented in an environment such as Github. Additionally, comparisons should be made using different sources. In this way, the success of the A2CPER method can be monitored.

The article can be reviewed again if the authors present their methods on a platform such as Github. That's why I'm requesting major revision.

Reviewer 4 ·

Basic reporting

no comment

Experimental design

no comment

Validity of the findings

no comment

Additional comments

Metaheuristic algorithm for strategy optimization of discrete problems is an emerging but promising research direction, which has far-reaching significance for some specific situations. In this paper, the authors also focused on this task and made a certain contribution, which is meaningful on the whole. But the reviewers still have some questions about the article that require the author to consider and elaborate, for instance:
1.Although many optimization algorithms are proposed with reference hyperparameters provided by the original authors, the values of these parameters may vary for different engineering problems. Therefore, for this paper, the hyperparameter selection of the proposed algorithm and the comparison algorithms needs to be further analyzed and experimentally validated.
2.Many metaheuristic algorithms proposed in recent years also have the potential to be extended to the field of discrete problem optimization, such as crayfish optimization algorithm (COA), remora optimization algorithm (ROA), and dynamic harris hawks optimization with mutation mechanism (DHHO/M). It is recommended to discuss this in the related work section to broaden the readership and enhance the advancement of the paper.
3.The key to solving practical engineering problems using optimization algorithms lies in mathematically modeling the problem and defining the optimization objective function. It is suggested that the author present a thorough reflection on the problems addressed in the paper to provide readers with a more comprehensive insight.
4.There is a lack of complexity analysis for the improved algorithm. The runtime and convergence speed are also alternative metrics regarding efficiency.
5.Recently, some general meta-heuristic algorithm improvement strategies, such as memory backtracking (Ref: memory backtracking strategy: an evolutionary updating mechanism for meta-heuristic algorithms), have been proposed. It is recommended that the authors apply these strategies to the proposed algorithm and validate their effectiveness on the engineering problems addressed in this paper.
6.Correct the grammatical mistakes and polish them with native speakers if possible.

·

Basic reporting

Clear and unambiguous, professional English is not used throughout this paper. Confusing, non-technical terms like 'denominated' and 'amalgamates' are used instead of clear professional technical terms like 'named' and 'used'. In addition overly descriptive, not technically evaluated adjectives are used in the paper making it difficult to read. For example, (1) 'the salient components of the A2CPER algorithm encompass several key facets:' should be 'The components of the A2CPER algorithm are:' and (2) 'for addressing intricate real-world tasks and decision-making quandaries' should be 'for addressing real-world tasks and decision-making'.

The literature references occur within the text in a format that reduces readability because there is no space between the names and year (i.e. Yoshimasa Kubo,Eric Chalmers,Artur Luczak.2022). Furthermore, references are entered twice in the bibliography and two different formats are used:

Yang Y, Gao W, Modares H,2021. et al. Robust actor–critic learning for continuous-time nonlinear systems with unmodeled dynamics[J]. IEEE Transactions on Fuzzy Systems, 2021, 30(6): 2101-2112.

Y. Yang, W. Gao, H. Modares and C. -Z. Xu,2022. "Robust Actor–Critic Learning for Continuous-Time Nonlinear Systems With Unmodeled Dynamics," in IEEE Transactions on Fuzzy Systems, vol. 30, no. 6, pp. 2101-2112, June 2022, doi:
577 10.1109/TFUZZ.2021.3075501.

To the extent sufficient field background context should be provided there are some missing references. The following papers all are recent papers related to using self attention mechanisms in deep learning and machine learning approaches and applying them to improve efficacy for discrete problems.

Ding, M., Han, C., & Guo, T. (2021). High generalization performance structured self-attention model for knapsack problem. Discrete Mathematics, Algorithms and Applications, 13(06), 2150076.

Wu, Z., Wang, S., Yuan, Q., Lou, N., Qiu, S., Bo, L., & Chen, X. (2023). Application of a deep learning-based discrete weather data continuouization model in ship route optimization. Ocean Engineering, 285, 115435.

Diallo, S. Y., Padilla, J., & Ezell, B. (2018). Assessing cyber-incidents using machine learning. International Journal of Information and Computer Security, 10(4), 341-360.

The mixed reference styles within the references section of the article are not professional. With respect to the figures: (a) it is unclear what Figure 1 is and how what it elucidates contributes to the paper, and (b) figure 9 relies on red/green color contrast. There are a nontrivial number of readers who suffer from red/green color blindness. Using a color-blindness safe color palette (https://davidmathlogic.com/colorblind/#%23D81B60-%231E88E5-%23FFC107-%23004D40) and larger text fonts would improve the readability of figures. In addition, while the source code for the algorithm is shared, the raw data, the source code for the other approaches and code used to generate the graphs in the paper has not been shared.

The paper includes self-contained research questions with relevant results to hypotheses.

Formal results should include clear definitions of all terms and theorems, and detailed proofs. However, in the paper it is unclear what the terms are in the equations. They are not adequately explained in the text or captioned.

Experimental design

The paper reflects original primary research within Aims and Scope of the journal. The research question relevant is meaningful. It is stated in a way that it is clear how research fills an identified knowledge gap. However, it does not define what a mearningful improvement for the A2CPER to achieve with respect to the other alternatives presented in the paper would be.

The investigation is rigorous investigation and performed to a high ethical standard. However, its technical standard needs to be improved. This related to the lack of what a meaningful improvement for the A2CPER to achieve with respect to the other alternatives presented in the paper would be. In any evaluation one approach will perform the best and one will perform the worst. However, a rigorous technical evaluation needs to demonstrate the the difference in performance for the best approach is materially different from the other approaches by testing the data and showing that the increase in performance is statistically significant.

While the source code for the algorithm is shared, the raw data, the source code for the other approaches and code used to generate the graphs in the paper has not been shared. These additional artifacts need to be provided to ensure replication of the results.

Validity of the findings

Novelty is assessed. But without testing for statistically significant improvement from A2CPER compared to the other approaches the impact cannot be assessed. The benefit to the literature is clearly stated but without the inclusion of the raw data and additional source code used in the evaluation and visualizition meaningful replication will not be achieved.

The underlying data have not been provided in any form. There are not tables presented in the paper. It is not possible to determine if they are robust, statistically sound, & controlled.

The conclusions are well stated, linked to original research question & limited to supporting results.

---

## Round 0.2 · Minor Revisions

The article still needs revision in terms of format and style.

Line 175 "attention-based" title is inadequate description. Please change this title to be more descriptive.
Change the title of "Cartpole-v1 Conclusions" in Line 702 to "Results of Cartpole-v1".
Change the title of "Acrobot-v1 Conclusions" in Line 782 to "Results of Acrobot-v1".
Change the title of "MountainCar-v0 Conclusions" in Line 837 to "Results of MountainCar-v0".
Change the "Summary" title in Line 864 to "Conclusions".
Change the title "Experimental of MountainCar-v0" in line 808 to "Experiments of MountainCar-v0".
Change the title of "Experimental of Acrobot-v1" in Line 731 to "Experiments of Acrobot-v1".
Change the title of "Experimental of CartPole-v1" in Line 614 to "Experiments of CartPole-v1".
Change the "Experiments" title in Line 502 to "Experimental setups and models".
Change the "Experimental setup CartPole-v1" title in Line 503 to "Experimental setup for CartPole-v1".
Change the "Experimental setup Acrobot-v1" title in Line 550 to "Experimental setup for Acrobot-v1".
Change the "Experimental setup MountainCar-v0" title in line 580 to "Experimental setup for MountainCar-v0".
Line 864: In the results section, the results of the research should be explained by saying "This study” or “In this study". It is not clear whether these results are the results of your article. So this section should be rearranged.

Reviewer 4 ·

Basic reporting

Most of the previous opinions have been revised, except for the following issues that need to be addressed by the author: the references cited by the author in the introduction are Catfish Optimization Algorithm (COA) and Raven Optimization Algorithm (ROA), while the references mentioned in the initial review comments are Crayfish Optimization Algorithm (COA) and Remora Optimization Algorithm (ROA). The author is requested to review relevant literature and make precise revisions.

Experimental design

no comment

Validity of the findings

no comment

Additional comments

no comment

·

Basic reporting

My basic reporting concerns from the previous review have been sufficiently addressed. The paper uses clear and unambiguous, professional English used throughout. The literature references, and sufficient field background/context are provided. It exhibits professional article structure, figures, tables. The raw data are shared. The results are relevant to the hypotheses and self-contained. There are clear definitions of all terms and theorems, and detailed proofs.

Experimental design

My experimental design concerns from the previous review have been sufficiently addressed. The paper exhibits original primary research within Aims and Scope of the journal. The research question is well defined, relevant & meaningful. It is stated how research fills an identified knowledge gap. The paper performs rigorous investigation to a high technical & ethical standard. The methods are described with sufficient detail & information to replicate.

Validity of the findings

My concerns related to the validity of the findings in the paper have been sufficiently addressed. There is meaningful replication and the rationale & benefit to literature is clearly stated. All underlying data have been provided; they are robust, statistically sound, & controlled. The conclusions are well stated, linked to original research question & limited to supporting results.

Additional comments

All my concerns have been sufficiently addressed. The paper is now suitable for publication.

---

## Round 0.3 · accepted · Accept

I am pleased to inform you that, following a thorough review of the revised version of your manuscript, it has been decided to accept it.

Reviewer 4 ·

Basic reporting

no comment

Experimental design

no comment

Validity of the findings

no comment

Additional comments

The article meets the PeerJ criteria and can be accepted.